

# Metagenomic analysis of nitrogen and methane cycling in the Arabian Sea oxygen minimum zone

Claudia Lüke[1,*], Daan R. Speth[1,*], Martine A.R. Kox[1], Laura Villanueva[2] and Mike S.M. Jetten[1,3,4]

[1] Department of Microbiology, IWWR, Radboud University Nijmegen, Nijmegen, Netherlands
[2] Department of Marine Organic Biogeochemistry, Royal Netherlands Institute for Sea Research (NIOZ), 't Horntje (Texel), Netherlands
[3] Department of Biotechnology, Delft University of Technology, Delft, Netherlands
[4] Soehngen Institute of Anaerobic Microbiology, Nijmegen, Netherlands
[*] These authors contributed equally to this work.

Corresponding authors
Claudia Lüke, clueke@science.ru.nl
Daan R. Speth, D.speth@science.ru.nl

## ABSTRACT

Oxygen minimum zones (OMZ) are areas in the global ocean where oxygen concentrations drop to below one percent. Low oxygen concentrations allow alternative respiration with nitrate and nitrite as electron acceptor to become prevalent in these areas, making them main contributors to oceanic nitrogen loss. The contribution of anammox and denitrification to nitrogen loss seems to vary in different OMZs. In the Arabian Sea, both processes were reported. Here, we performed a metagenomics study of the upper and core zone of the Arabian Sea OMZ, to provide a comprehensive overview of the genetic potential for nitrogen and methane cycling. We propose that aerobic ammonium oxidation is carried out by a diverse community of *Thaumarchaeota* in the upper zone of the OMZ, whereas a low diversity of *Scalindua*-like anammox bacteria contribute significantly to nitrogen loss in the core zone. Aerobic nitrite oxidation in the OMZ seems to be performed by *Nitrospina spp.* and a novel lineage of nitrite oxidizing organisms that is present in roughly equal abundance as *Nitrospina*. Dissimilatory nitrate reduction to ammonia (DNRA) can be carried out by yet unknown microorganisms harbouring a divergent *nrfA* gene. The metagenomes do not provide conclusive evidence for active methane cycling; however, a low abundance of novel alkane monooxygenase diversity was detected. Taken together, our approach confirmed the genomic potential for an active nitrogen cycle in the Arabian Sea and allowed detection of hitherto overlooked lineages of carbon and nitrogen cycle bacteria.

## INTRODUCTION

Oxygen is a key parameter for biogeochemical cycling and has major impact on the marine nitrogen and carbon turnover. The vast majority of the global ocean waters is well oxygenated, allowing aerobic micro- and macro-organisms to thrive. However, in several areas, underlying regions of high productivity, dissolved oxygen concentrations drop to very low levels. These regions are referred to as oxygen minimum zones (OMZ). There is

not a general agreement on the definition of an OMZ; however, an oxygen concentration of ≤20 µM was proposed (*Lam & Kuypers, 2011*). Using this threshold, approximately 1% of the total ocean volume can be defined as an OMZ (*Lam & Kuypers, 2011*). In the eastern tropical North Pacific (ETNP), the eastern tropical South Pacific (ETSP) and the Arabian Sea, the three prominent OMZs, oxygen concentration can even drop below levels detectable by sensitive modern techniques (*Revsbech et al., 2009*; *Thamdrup, Dalsgaard & Revsbech, 2012*).

Despite comprising only a small fraction of the total ocean volume, OMZs contribute 30–50% of the nitrogen loss from the ocean (*Gruber & Sarmiento, 1997*; *Codispoti et al., 2001*). This can be attributed to a highly active nitrogen cycle in these systems (*Lam & Kuypers, 2011*). After depletion of oxygen, nitrate is the next most energetically favourable terminal electron acceptor and is present in micro-molar concentrations in OMZs (e.g., *Pitcher et al., 2011*). Nitrate reduction coupled to the oxidation of organic matter releases 16 mole ammonium per mole organic matter oxidized (*Redfield, Ketchum & Richards, 1963*). In addition to this, ammonium can be produced by dissimilatory nitrite reduction to ammonium (DNRA). Despite being aerobic processes, ammonium and nitrite oxidation occur in OMZs, partially converting the ammonium back to nitrite and nitrate (*Kalvelage et al., 2011*; *Kalvelage et al., 2015*; *Füssel et al., 2012*). Eventually, nitrogen is lost from the system due to denitrification (*Groffman et al., 2006*; *Ward et al., 2009*) or anaerobic ammonium oxidation (anammox) (*Kuypers et al., 2003*; *Kalvelage et al., 2015*).

The relative contribution of anammox and denitrification to nitrogen loss from OMZs has been the subject of debate (*Lam & Kuypers, 2011*). Before the discovery of anammox, denitrification was thought to be the only contributor (*Codispoti & Richards, 1976*; *Lipschultz et al., 1990*; *Devol et al., 2006*), but the detected ammonium concentrations were lower than expected based on just denitrification. After the discovery of the anammox process in the Black Sea (*Kuypers et al., 2003*), anammox bacteria were shown to occur in all the major OMZ using marker genes, lipid analysis, FISH and stable isotope pairing (e.g., *Kuypers et al., 2005*; *Thamdrup et al., 2006*; *Hamersley et al., 2007*). In the ETSP OMZ, anammox was the dominant process involved in nitrogen loss (*Lam & Kuypers, 2011*; *Stewart, Ulloa & DeLong, 2012*). For the Arabian Sea OMZ, evidence for both denitrification and anammox as the dominant cause of nitrogen loss exists (*Ward et al., 2009*; *Jensen et al., 2011*; *Pitcher et al., 2011*) and the contribution of either process likely varies with season and location.

Previous studies on nitrogen cycling the Arabian Sea OMZ focused on one or (a comparison of) a few processes such as ammonia oxidation (*Newell et al., 2011*; *Schouten et al., 2012*), denitrification (*Jayakumar et al., 2004*; *Ward et al., 2009*), and anammox (*Jensen et al., 2011*; *Pitcher et al., 2011*; *Villanueva et al., 2014*). A recent study used metagenomes obtained from samples across the ETSP, ETNP and the Bermuda Atlantic Time-series Station to access the distribution of iron and copper containing nitrogen cycle enzymes in these systems (*Glass et al., 2015*), but a comprehensive study of the nitrogen cycling potential in the Arabian Sea OMZ is lacking.

Furthermore, only very little is known on methane turnover in oxygen minimum zones. Low concentrations of methane have been reported at sites of the ETNP and the

**Table 1  Overview of nitrogen and methane cycle marker genes and BLAST score ratio cut-off value used for removal of false positive BLAST hits.** See Fig. S2 for a graphical overview of the bit-score ratio analysis pipeline. DNRA: dissimilatory nitrite reduction to ammonia. Anammox: anaerobic ammonia oxidation.

| Process | Enzyme name | Gene abbreviation | BLAST score ratio cutoff (PA2/PA5) |
|---|---|---|---|
| Nitrogen fixation | Nitrogenase | *nifH* | #/# |
| Nitrification | Ammonium monooxygenase | *amoA* | #/# |
| | Hydroxylamine oxidoreductase | *hao* | # / 0.75 |
| | Nitrate:nitrite oxidoreductase | *nxrA* | 0.85*/0.85* |
| Denitrification/DNRA/anammox | Nitrate reductase | *narG* | 0.5/0.5 |
| Denitrification | Copper nitrite reductase | *nirK* | 0.55/0.55 |
| | Heme *cd1* nitrite reductase | *nirS* | #/0.6 |
| | Nitric oxide reductase | *norB/norZ* | 0.8/0.8 |
| | Nitrous oxide reductase | *nosZ* | 0.8/0.75 |
| DNRA | Cytochrome c nitrite reductase | *nrfA* | #/# |
| Anammox | Hydrazine synthase | *hzsA* | #/0.75 |
| | Hydrazine dehydrogenase | *hdh* | **/** |
| Methanogenesis/anaerobic methane oxidation | Methyl-coenzyme M reductase | *mcrA* | #/# |
| Aerobic methane oxidation | Soluble methane monooxygenase | *mmoX* | #/# |
| | Particulate methane monooxygenase | *pmoA* | #/# |
| Methylphosphonate production | Methylphosphonate synthase | *mpnS* | #/# |
| Methylphosphonate cleavage | C-P lyase | *phnGHI* | #/# |

**Notes.**
[#] Manually checked.
[*] Subset of *narG* hits.
[**] Subset of hao hits after mapping against Scalindua *hdh*.

ETSP suggesting the possibility of methane cycling in these ecosystems (*Sansone & Popp, 2001*; *Padilla et al., 2016*). In addition, *pmoA* genes (encoding the particulate methane monooxygenase, see Table 1) have been detected in the water column of the ETSP OMZ indicating the presence of aerobic methanotrophs (*Tavormina et al., 2013*). In the Arabian Sea, high methane concentrations (up to 227% saturation compared to atmospheric levels) have been measured in the surface waters (*Owens et al., 1991*; *Bange et al., 1998*; *Upstill-Goddard, Barnes & Owens, 1999*) and elevated concentrations (up to approximately 8 nM) were found at 150–200 m depths (*Jayakumar et al., 2001*). Not much is known about the processes and the organisms involved in production or turnover, but recently *Thaumarchaeota* cleaving methylphosphonates have been proposed as a potential source of methane in the oceans (*Metcalf et al., 2012*).

Metagenomics is a powerful tool to provide an all-inclusive picture of the functional potential of an ecosystem. As sequencing is becoming easier and less expensive, the bottleneck in metagenomics is shifting from data generation to sequence analysis strategies. Here, we developed a new analysis strategy for mining metagenome data, based on curated databases of marker genes for nitrogen and methane cycle processes (see Table 1 for an overview of the used marker genes). We applied this strategy to metagenome data retrieved from two depths along the oxygen gradient of the Arabian Sea OMZ.

## MATERIALS AND METHODS

### Sampling and sample preparation

Samples used in this manuscript were collected in the Arabian Sea OMZ during the PASOM cruise funded by the Netherlands Organization for Scientific Research (NWO) under number 817.01.015. The sampling site and sampling procedure are described in detail in a previous study (*Pitcher et al., 2011*). Briefly, the site is located within the Arabian Sea (lat. 21°55.6′, long. 63°10.6′) and influenced by Persian Gulf Water and Red Sea Water. It is just outside the region characterized by a quasi-permanent secondary nitrite maximum (*Revsbech et al., 2009*). Twelve samples were taken along a depth profile (stations PA1-PA12) in the water column ranging from the ocean surface (0 m) down to 2,000 m below the surface (total water depth at this site is 3,010 m). Dissolved oxygen concentrations decreased from fully saturated concentrations at the surface to 3.2 µM at the core of the OMZ. The latter value reflects the detection limit of the CTD oxygen sensor, the real values are probably substantially lower as suggested by STOX (Switchable Trace amount OXygen; *Revsbech et al., 2009*) sensor measurements in the Arabian Sea OMZ, where reported values were ≤0.09 µM which was the detection limit of the sensor in that sampling campaign (*Jensen et al., 2011*).

In our sampling dataset, the depth of 170 m below ocean surface was defined as OMZ transition zone whereas the depth of 600 m belongs to the OMZ core zone at this sampling site (*Pitcher et al., 2011*). Oxygen concentrations increased again at 1,050 m (*Pitcher et al., 2011*). At 170 m depth, the ammonium concentration showed a peak (0.14 µM), nitrite concentrations had peaks at 170 m and 600 m below surface (*Pitcher et al., 2011*). Furthermore, biomarker analysis based on the distribution of lipids and copy numbers of the 16S rRNA/*amoA* (encoding the ammonia monooxygenase)/*hdh* (encoding the hydrazine dehydrogenase) genes and/or transcripts indicated highest numbers of *Thaumarchaeota* at 170 m and highest numbers of anammox at 600 m depth (*Pitcher et al., 2011*). Thus, in this study we focused on two different depths in the water column: 170 m below ocean surface (station PA2) and 600 m below ocean surface (station PA5). Main physiochemical parameters were taken from *Pitcher et al. (2011)* and are reported in Table S1. From these depths, large-volumes of seawater (200–1,700 L) were filtered through 142-mm diameter 0.2 µm polycarbonate filters (Millipore, Billerica, MA). Filters were cut into fragments and DNA extraction was performed as described by Pitcher and co-workers (*Pitcher et al., 2011*). After extraction, DNA was precipitated using ice-cold ethanol, dried, and re-dissolved in 100 µl of 10 mM Tris–HCl, pH 8. Total nucleic acid concentrations were quantified spectrophotometrically (Nanodrop; Thermo Scientific, Wilmington, DE, USA) and checked by agarose gel electrophoresis for quality. Extracts were kept frozen at −80 °C.

### Ion Torrent library preparation and sequencing

All kits used in this section were obtained from Life technologies (Life Technologies, Carlsbad, CA, USA). For both samples an identical library preparation was performed. Genomic DNA was sheared for 7 min using the Ion Xpress™ Plus Fragment Library Kit following the manufacturer's instructions. Further library preparation was performed using the Ion Plus Fragment Library Kit following manufacturer's instructions. Size selection of

the library was performed using an E-gel® 2% agarose gel, resulting in a median fragment size of approximately 330 bp. Emulsion PCR was performed using the OneTouch 200 bp kit and sequencing was performed on an Ion Torrent PGM using the Ion PGM 200 bp sequencing kit and an Ion 318 chip, resulting in 4.9 million reads for station PA2 and 2.6 million reads for station PA5. The raw reads were submitted to the NCBI Sequence Read Archive under the accession number SRA304624.

## Bioinformatics

### Quality assessment and assembly feasibility

Raw sequence reads were imported into the CLC Genomics Workbench (v7.0.3, CLCbio Arhus, Denmark) and end-trimmed on quality using the CLC genomics default settings (quality limit 0.05 and two ambiguous nucleotides allowed) and length ($\geq$100 bp) resulting in 3.3 million reads of station PA2 and 1.6 million reads from station PA5, which were used for subsequent analyses.

To assess the feasibility of assembly for the analysis of the read data, the datasets were assembled *de novo* using the CLC genomics workbench with word size 35 and bubble size 5,000. Using the data from both sampling sites combined, this resulted in 5,105 contigs longer than 1,000 bp, incorporating only 7.6% of the reads. Based on the results of the metagenome assembly, we decided to employ a read based analysis strategy.

### 16S rRNA gene analysis

To extract reads matching the ribosomal small subunit for taxonomic classification, the SILVA SSU RefNR99 dataset (version 115, *Quast et al., 2013*) was used as reference. First, the metagenome reads were mapped against the SILVA dataset using the CLC Genomics Workbench (mismatch penalty 2, In/Del penalty 3, 50% identity over 70% of the sequence). Mapped reads were extracted and used for BLAST searches against the identical SILVA dataset ($E$-value cut-off $10^{-6}$). Positive hits were aligned using the SINA aligner (*Pruesse, Peplies & Glockner, 2012*) and imported into the SILVA refNR99 version 115 ARB database (*Ludwig et al., 2004*; *Quast et al., 2013*). Sequences were added to the existing Neighbor Joining 16S rRNA tree (including nearly 500,000 16S rRNA sequences) using maximum parsimony criteria without optimization of the tree topology (ARB parsimony Quick add marked). All phylogenetic clusters containing reads from our metagenomes were visually inspected and the number of reads in the respective cluster was recorded. For reads that did not cluster unambiguously to a specific group, the phylogeny of the higher taxonomic level was recorded.

### Functional gene analysis

*Construction of reference datasets.* To screen the metagenomes for potential function, reference datasets of marker genes (amino acid sequences) were manually curated. Marker genes for nitrogen cycle processes (nitrogen fixation, nitrification, denitrification, dissimilatory nitrite reduction to ammonia (DNRA) and anammox) and marker genes for methane cycle processes (methanogenesis, methane oxidation, methylphosphonate turnover) were selected. Table 1 gives an overview of genes and gene products used in this study. Reference databases were curated for all genes individually. Where available,

existing databases were used as a reference, as for the *nifH* gene (*Gaby & Buckley, 2014*), the bacterial *amoA*/*pmoA* gene (*Dumont et al., 2014*), the archaeal *amoA* gene (*Pester et al., 2012*) and the *mcrA* gene (*Angel, Matthies & Conrad, 2011*). From these ARB databases, a subset (490 *nifH* sequences, 167 bacterial *pmoA*/*amoA* sequences also including novel clades from a manuscript in revision (all sequences are provided in the Supplemental Information), 134 archaeal *amoA* sequences, 124 methanogen *mcrA* sequences) covering the main described clusters was selected from the phylogentic trees in ARB and extracted for the analysis in this study. For the *mcrA* gene, 78 sequences from described methanotrophic ANME clusters (*Knittel & Boetius, 2009*) were added to the selection. In addition, *mcrA* sequences from *Bathyarchaeota* (*Evans et al., 2015*) were downloaded from NCBI. For the remaining genes (*hao*/*hdh*, *narG*/*nxrA*, *nirS*, *nirK*, *nosZ*, *nrfA*, *hzsA*, *mmoX*, *mpnS*, *phnGHI*), sequences were collected from public databases (proteins that matched to the respective Interpro family and keyword/accession number search in NCBI). The amino acid sequences were downloaded and aligned using Clustal Omega (*Sievers et al., 2011*). Alignments were manually inspected and for sequences with large gaps or insertions, literature was queried to verify the function of the respective enzyme. In cases where the function of a divergent sequence has not been described or is different than the target database, the sequence was discarded. The remaining aligned sequences were imported into ARB (*Ludwig et al., 2004*). Sequences with ambiguous characters or obvious shifts in reading frame were removed. The remaining sequences were used for phylogenetic tree construction. Phylogenies were compared to literature to verify that the known and described phylogenetic groups are covered and that the dataset is representative. For the *norB*/*norZ* database, all sequences in the cytochrome oxidase PFAM family (PF00115) were downloaded, replicates were removed using UClust (80% identity) (*Edgar, 2010*) and the clustered sequences were screened against characterized cNOR (cytochrome c dependent nitric oxide reductase) and qNOR (quinol dependent nitric oxide reductase) sequences using a BLAST score ratio (BSR) (*Rasko, Myers & Ravel, 2005*). Sequences with a BSR over 0.4 were included in the database. Representatives of phylogenetic clusters (distance level of 0.1) were exported in fasta format and imported into ARB where sequences with ambiguous characters or obvious shifts in reading frame were removed. The remaining sequences were used for phylogenetic tree construction and exported as reference set for subsequent BLAST analysis. All reference sets are provided as Supplemental Information.

*BLAST analysis using the reference datasets.* The reads of the metagenomes described above were used in BLASTx searches against the reference sets ($E$-value cutoff $10^{-6}$). The $E$-value was established after first test analyses with different $E$-values ($10^{-4}$–$10^{-6}$). Positive hits were extracted using a custom perl script available at www.github.com/dspeth. In a second step, positive reads were used in BLASTx searches against the NCBI NR database. For reference datasets resulting in low number of target sequences (approximately $n < 100$), false positive reads were removed by manual inspection of the BLAST results. For large number of target reads (approx. $n > 100$), a modified BLAST score ratio (BSR) approach was used to remove false positive hits while keeping divergent sequences. In this approach, rather than calculating the ratio between the score of a hit against our database and the

maximum score (of a self-hit) for every read (*Rasko, Myers & Ravel, 2005*), we calculated the ratio between a hit against our database and a hit against the NCBI-NR database (NCBI non-redundant protein sequence database, see Fig. S2 for the procedure overview). This guaranteed that sequences with low similarity to both NR and our database were kept as true positives, whereas reads with a much better hit to the NR than to our database were discarded as false positives. We determined a separate threshold value for a positive hit for each specific gene database (Table 1). This threshold was selected based on a first manual inspection of reads with different bit-score ratios (Figs. S3–S7). To distinguish between the closely related *hao* (encoding the hydroxylamine oxidoreductase) and *hdh* (encoding the hydrazine dehydrogenase) sequences and the *narG* (encoding the nitrate reductase) and *nxrA* (encoding the nitrite oxidoreductase) sequences respectively, a second step was applied. Positive *hao* reads obtained after the BSR analysis were mapped in CLC genomics (mismatch penalty 2, In/Del penalty 3, minimum 50% identity over 50% of the read) against both copies of the *Scalindua brodae hdh* (accession numbers: KHE92657.1, KHE91265.1). Mapped reads were extracted and classified as *hdh*, non-mapped reads were classified as *hao*. To distinguish between *narG* and *nxrA*, positive *narG* reads obtained after the first BSR analysis were used in a second round of BLASTx against two *nxrA* gene subsets, one containing the *Nitrobacter/Nitrococcus/Nitrolancea nxrA* sequences and the other containing the *Nitrospira/Nitrospina/* anammox *nxrA* sequences. A BSR cut-off of 0.85 was used to separate *nxrA* from *narG*.

For all genes, after removal of false positive targets using the BSR analysis, reads were assigned to taxonomy using MEGAN (*Huson et al., 2007*; *Huson et al., 2011*) and the taxonomy of maximum 5 hits with over 90% score from the top hit.

### Normalization of metagenomic read counts

After phylogenetic assignment, the positive read counts were normalized according to gene length and metagenome size according to the following procedure: (Number of positive reads)/((length of gene in bp) * (number of total reads in the metagenome)). The *rpoB* gene (encoding the RNA polymerase) was used as a single copy gene reference. To estimate the abundance of microorganisms encoding nitrogen or methane cycling marker genes in their genomes, the normalized read counts were discussed as fraction of the normalized total read counts of the *rpoB* gene.

### Assembly of selected reads

For selected genes (*hdh, hzsA, nirS, nrfA, nxrA*), the reads were extracted and imported in the CLC genomics workbench to reconstruct (near) full-length genes. In all cases, reconstruction through direct assembly (word size 35, bubble size 5,000) was attempted. This was successful for *hzsA, nirS* and *nrfA*, where diversity was limited. For the more diverse data (*nxrA* and *hdh*) iterative mapping (*Dutilh, Huynen & Strous, 2009*) was used. *Hdh* could be reconstructed by mapping against both copies of *Scalindua brodae hdh* (KHE92657.1 & KHE91265.1; mismatch penalty 2, In/Del penalty 3, minimum 50% identity over 50% of the read). The *nxrA* sequences of *Nitrospina sp.* and *Scalindua sp.* were reconstructed by iterative mapping on the sequences of *Nitrospina gracilis* and *Scalindua brodae* (WP_042250442.1 & KHE93157.1; mismatch penalty 2, In/Del penalty 3, minimum

80% identity over 50% of the read). Novel *nxrA* sequences were retrieved using iterative mapping of the reads that could not be assigned to either *Nitrospina* or *Scalindua* on the *nxrA* sequence of *Kuenenia stuttgartiensis* (CAJ72445.1; mismatch penalty 1, In/Del penalty 3, minimum 30% identity over 50% of the read).

## RESULTS AND DISCUSSION

Metagenomics can be used as powerful tool to gain insights into the functional potential of an ecosystem. As the sequencing procedure itself is becoming easier and less expensive resulting in generation of large amounts of data, sequence analysis strategies are becoming a bottleneck in time and resources. Depending on the diversity and complexity of the dataset, different analysis approaches are needed. In this study, we provide a strategy for the systematic screening of metagenomes for nitrogen and methane cycling potential using curated functional gene reference databases. We applied our strategy to a dataset from the Arabian Sea oxygen minimum zone (OMZ) analysing the genetic potential for nitrogen and methane turnover in the upper limit (station PA2) characterized by low oxygen (approximately 5 μM, Table S1) and the core zone (station PA5) in which the oxygen concentration drops below the detection limit (3.4 μM) (Table S1).

### Taxonomy based on 16S rRNA gene analysis

To get insight into the overall microbial community in the Arabian Sea OMZ, we analysed 16S rRNA gene reads retrieved from the metagenomes at station PA2 and PA5 (Fig. 1). At the upper limit of the OMZ (station PA2, Table S1), the SAR11 clade (*Alphaproteobacteria*) formed one of the most abundant microbial groups (14%). This clade represents in general one of the most abundant microorganisms in seawater, contributing up to 30% of all bacterioplankton (*Morris et al., 2005*). Sub-clusters within the SAR11 clade have been linked to ecotypes occupying different niches in the ocean water column (*Field et al., 1997*; *Vergin et al., 2013*). In the PA2 dataset, most sequence reads clustered within SAR11 subgroup 1, closely related to cultivated strains of 'Candidatus Pelagibacter ubique.' 'Ca. P. ubique' has a small, streamlined genome adapted to rapid heterotrophic growth (*Rappe et al., 2002*; *Giovannoni et al., 2005*) and is unlikely to directly contribute to nitrogen cycling in the Arabian Sea OMZ. The remainder of the SAR11 reads were distributed across the entire SAR11 clade. Ammonium oxidizing *Archaea* (AOA) of the Marine Group I (MG-I) *Thaumarchaeota* were as abundant as SAR11 at station PA2 (14%), confirming previous PCR- and lipid-based analyses showing that *Thaumarchaeota* were abundant at this location (*Pitcher et al., 2011*). Both SAR11 and MG-I were less abundant in the OMZ core, but still have a substantial presence of 8.1% and 3.3% respectively (Fig. 1). Reads affiliated with the bacterial SAR86 clade (*Gammaproteobacteria*) and archaeal Marine Group II (MG-II) made up 8% and 7% of the 16S rRNA gene reads at station PA2, but were only marginally present in the OMZ core (1.5 % and 0.4 % of the reads, respectively). This is consistent with an aerobic heterotrophic lifestyle predicted from previously obtained genomes of organisms of both lineages (*Dupont et al., 2012*; *Iverson et al., 2012*). So far, no metabolic adaptations of these organisms to an anaerobic lifestyle have been characterized. Nevertheless, the persistence, albeit in low abundance, of aerobic organisms in the anoxic OMZ core might

| Taxonomy | | | | PA2 | PA5 |
|---|---|---|---|---|---|
| **Bacteria** | Proteobacteria | Alpha | SAR11 | 14.3 | 8.1 |
| | | | Rhodobacteriacaea | 1.5 | 0.5 |
| | | | Rhodospirillales | 1.4 | 4.2 |
| | | Gamma | Alteromonadales | 1.9 | 1.1 |
| | | | SAR86 | 7.9 | 1.5 |
| | | | SUP05 and alike | 2.9 | 2.7 |
| | | | Salinisphaeraceae | 1.6 | 1.9 |
| | | | E01-9C-26 | 0.5 | 1.6 |
| | | Delta | SAR324 | 5.0 | 3.1 |
| | Bacteriodetes | | Flavobacteriales | 2.0 | 0.5 |
| | Planctomycetes | | Brocadiaceae | < 0.1 | 5.2 |
| | | | Phycisphaerae | < 0.1 | 1.1 |
| | Verrucomicrobia | | Arctic97B-4 | 1.1 | 0.2 |
| | Deferribacteres | | SAR406 | 8.5 | 19.9 |
| | Nitrospinae | | Nitrospina | 1.4 | 2.7 |
| | Actinobacteria | | Acidimicrobia | 3.2 | 2.4 |
| | Chloroflexi | | SAR202 | 0.8 | 2.4 |
| | | | Dehalococcoidia | 0 | 1.6 |
| **Archaea** | Thaumarchaeota | | Marine Group I | 14.0 | 3.3 |
| | Euryarchaeota | | Marine Group II | 7.5 | 0.4 |
| | | | Marine Group III | 1.4 | 0.5 |
| | | | Haloarchaea | 0 | 3.8 |
| Sum of remaining rare taxa | | | | 23.0 | 27.6 |

**Figure 1** **Overview of microbial 16S rRNA gene diversity in the Arabian Sea oxygen minimum zone.** Overview of microbial 16S rRNA gene diversity detected in the suboxic zone (station PA2) and the anoxic core (station PA5) of the Arabian Sea oxygen minimum zone. Sequence reads are shown as percentage of total 16S rRNA read counts. Only phylogenetic groups accounting for more than 1% of the total community in at least one of the two datasets are listed.

be partially explained by the attachment to slowly sinking organic particles, also referred to as marine snow (*Wright, Konwar & Hallam, 2012*). Formed in the metabolically active photic zone, these particles continuously sink through the water column to the floor of the ocean and thereby also passing the anoxic core of the OMZ. Association to particles has been previously proposed for SAR11 (*Zeigler Allen et al., 2012*) and demonstrated for the MG-II *Archaea* (*Orsi et al., 2015*).

The deep-branching bacterial phylum *Marinimicrobia* (formerly SAR406) (*Fuhrman, McCallum & Davis, 1993*; *Gordon & Giovannoni, 1996*; *Rinke et al., 2013*) comprised 9% of all 16S rRNA affiliated sequence reads at station PA2 and 20% at PA5. A recent transcriptome study indicates the involvement of these bacteria in extracellular proteolysis and fermentative amino acid degradation in a methanogenic environment (*Nobu et al., 2015*). These findings of an anaerobic lifestyle agree well with their high abundance in a low oxygen environment (upper OMZ) and their dominance in the OMZ core. The detected *Rhodospirillales* (*Alphaproteobacteria*) might also contribute to fermentation of organic matter, as this order comprises known acetic acid bacteria. *Rhodospirillales* were barely detectable at the OMZ upper limit, but comprise 4.1% of the community at the OMZ core. Other abundant lineages include the *Deltaproteobacteria* SAR324. These bacteria are frequently found in the ocean and seem to be correlated with low oxygen concentrations (*Wright, Konwar & Hallam, 2012*; *Sheik, Jain & Dick, 2014*). SAR324 representatives have been shown to be able to fix $CO_2$ (*Swan et al., 2011*) and are predicted to be capable of autotrophic denitrification with various electron donors (*Sheik, Jain & Dick, 2014*). Likewise, *Gammaproteobacteria* of the SUP05 lineage have been implied in autotrophic denitrification with sulphur as electron donor (*Lavik et al., 2009*; *Walsh et al., 2009*; *Russ et al., 2014*). They furthermore have been associated with cryptic sulfur cycling in the ETSP OMZ (*Canfield et al., 2010*). Members of both SAR324 and SUP05 have a similar presence at both stations (5% in PA2 and 3.1% in PA5 for the SAR324 lineage, 2.9% and 2.7% for the SUP05 lineage), with SAR324 slightly decreasing in the OMZ core (Fig. 1).

Anammox 16S rRNA genes are barely detected at the upper limit of the OMZ, consistent with previous analyses (*Pitcher et al., 2011*; *Villanueva et al., 2014*), but comprise almost 5% of the community at the OMZ core. Interestingly, a recent study showed that autotrophic denitrifiers oxidizing hydrogen sulphide could form a stable community with anammox bacteria in a reactor system (*Russ et al., 2014*). Whether this is also the case at the OMZ core remains to be investigated. Although some of the lineages discussed above are abundant, assembly of the complete metagenome reads did not yield any contigs with high sequencing depth, indicating the diversity within each lineage is substantial.

To assess the diversity and phylogeny of the detected AOA and anammox bacteria in more detail, reads matching the 16S rRNA gene of either group were extracted and assembled into contig sequences for phylogenetic tree construction. Reads affiliated with MG-I AOA from the OMZ upper limit (PA2) could be assembled into two representative contigs: Contig-1 comprises 41% of all extracted MG-I reads and contig-2 was built from 15% of the reads. Thus, the two contigs represent the majority of the thaumarchaeotal community. Nevertheless, these sequences do not represent a single genotype but represent a hybrid of 16S rRNA reads from multiple closely related organisms (Fig. S1). Contig-1

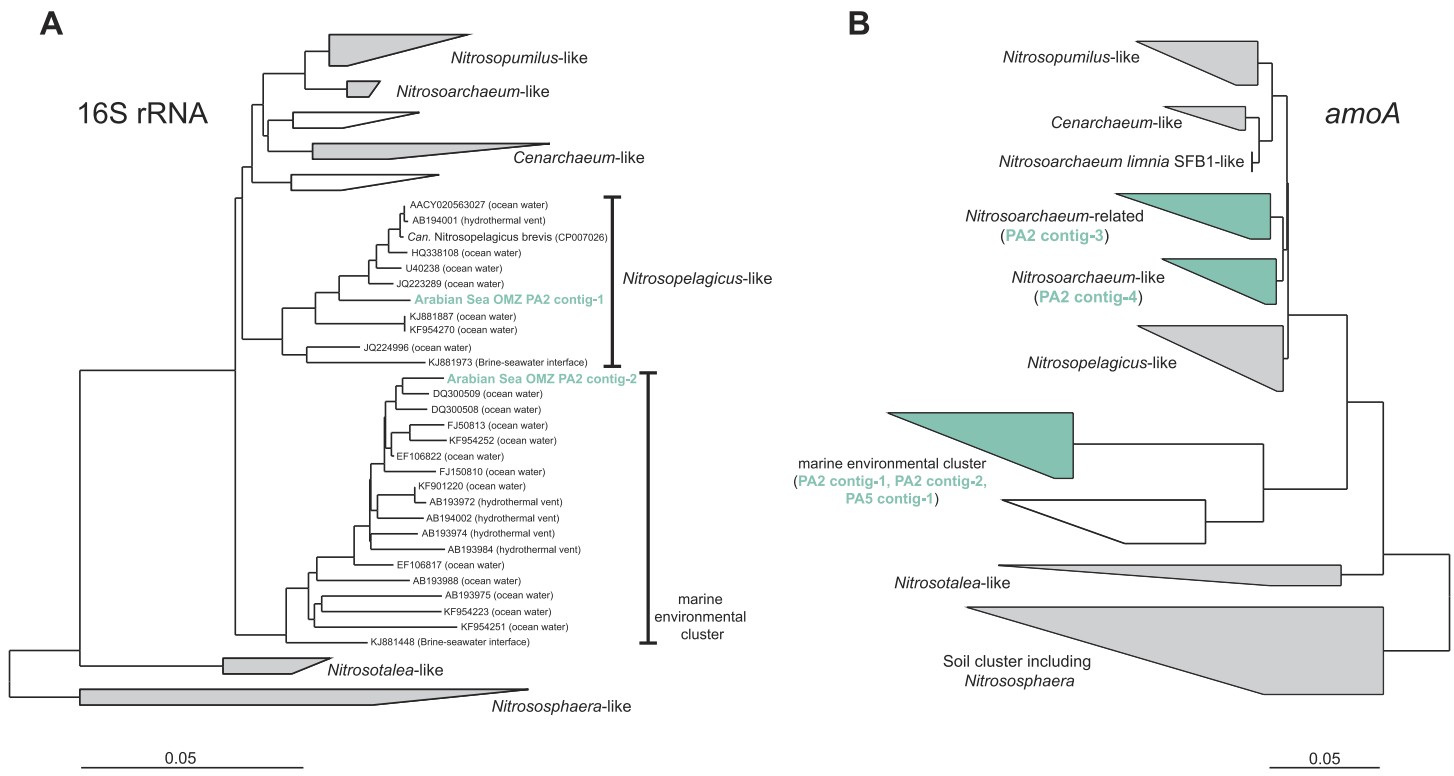

**Figure 2** **Phylogenetic inference of thaumarchaeal contigs assembled from the OMZ metagenomes.** The trees were calculated using the Neighbor Joining algorithm and based on 1,225 nucleotide positions for the 16S rRNA gene (A) and 144 deduced amino acid positions for the ammonia monooxygenase encoding gene (*amoA*) (B).

shows 98% identity to the very recently described '*Candidatus* Nitrosopelagicus brevis' (Fig. 2) (*Santoro et al., 2015*), contig-2 has only moderate identities to isolated or enriched ammonia oxidizers (93% identity to *Nitrosoarchaeum limnia* SFB1) (Fig. 2).

Reads from station PA5 clustering with the *Brocadiaceae*, the family comprising all known anammox bacteria, were extracted and could be assembled into one representative sequence. Here, diversity was considerably less than for the *Thaumarchaeota* and the contig represents a single genotype that is 96% identical to *Scalindua brodae*, the closest sequenced relative (*Speth et al., 2015*), and 99% identical to '*Candidatus* Scalindua arabica' clones previously obtained from the Arabian sea OMZ (Fig. 3) (*Woebken et al., 2008*). Interestingly, sequences obtained by Ward and co-workers from the Arabian Sea OMZ share only 97% sequence identity to the extracted contig (*Ward et al., 2009*) indicating a spatial or temporal niche differentiation of different *Scalindua*-like ecotypes in this system.

## Genetic potential for nitrogen and methane cycling

To assess the nitrogen and methane cycling potential in the metagenome of both the upper limit and the core of the OMZ, we performed BLASTx searches of the metagenomic reads against curated databases of key genes (Table 1) involved in nitrogen and methane cycle processes. To remove false positive hits while keeping divergent sequences, we used a modified BLAST score ratio (BSR) approach (see methods section).

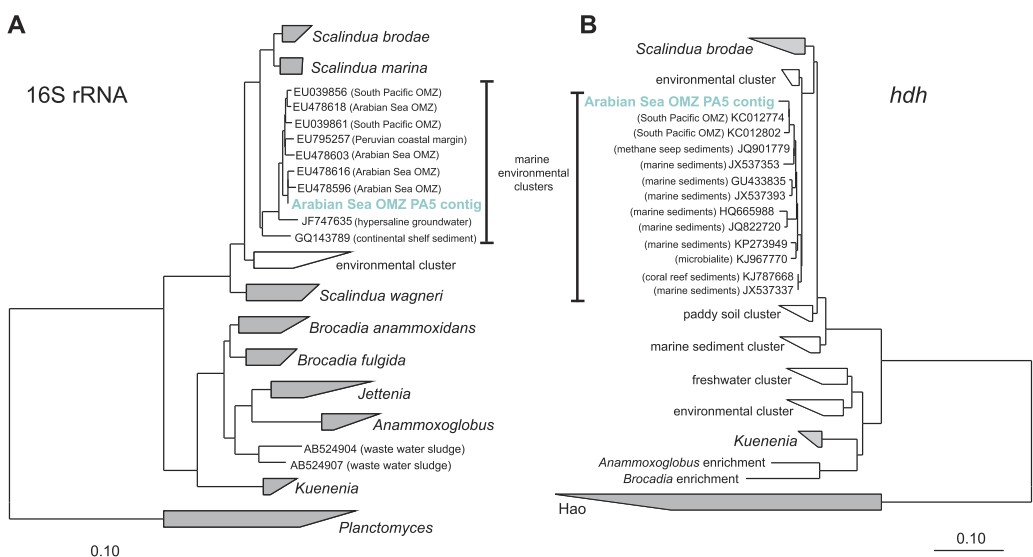

**Figure 3** **Phylogenetic inference of Scalindua-related contigs assembled from the OMZ metagenomes.** The tree were calculated using the Neighbor Joining algorithm and based on 1,388 nucleotide positions for the 16S rRNA gene (A) and 180 deduced amino acid positions for the hydrazine dehydrogenase encoding gene (*hdh*) (B).

## Nitrogen cycling potential
### *Nitrification*

Marker genes indicative for the first step in nitrification, the conversion from ammonia to nitrite, are the *amoA* (encoding a subunit of the membrane bound ammonium monooxygenase) and the *hao* gene (encoding the hydroxylamine oxidoreductase). We found 228 reads matching *amoA* in the PA2 dataset, 227 of which could be assigned to ammonia oxidizing Archaea (AOA), indicating that they contribute approximately 25% to the total microbial community in this sample (Fig. 4). This estimate exceeds the estimated abundance based on 16S rRNA genes (Fig. 1). Our analysis strategy includes the correction for sequencing depth and gene length, however, no correction for gene copy numbers was applied. This information can only be deduced from genomes and is not known for the vast majority of microorganisms. In the presence of genomes harbouring multiple rRNA operons, the total number of detected organisms is artificially inflated, leading to an underestimation of organisms with a single rRNA operon which would explain the lower 16S rRNA estimates in this dataset. An alternative, albeit less likely, explanation for higher *amoA* estimates can be the presence of multiple copies of the *amoA* gene in the detected AOA genomes, but this has not been observed in any previously sequenced AOA species.

The *amoA* reads could be assembled into 5 major contigs (4 contigs from site PA2 and 1 contig from PA5) that were compared to 16S rRNA phylogeny (Fig. 1). As for the 16S rRNA contig, *amoA* sequences could not be affiliated with a single MG-I species, but showed a diversity of at least 2 major genotypes. Unlike the 16S rRNA gene analysis, none of the contigs clustered with *Nitrosopelagicus*-like *amoA* sequences. Instead two out of the four obtained contigs from PA2 did not cluster with, but between the *Nitrosopelagicus*

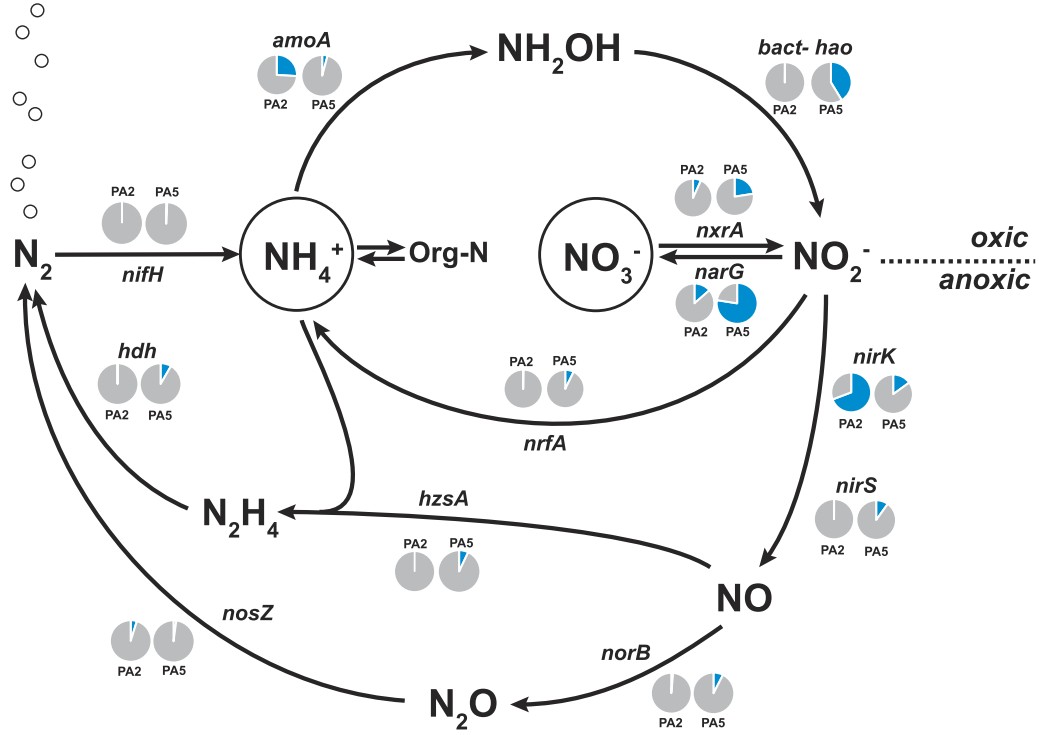

**Figure 4** **Nitrogen cycling potential in the Arabian Sea oxygen minimum zone.** Read abundances were normalized according to gene length and total read abundance in the metagenome dataset. Normalized abundances are shown as proportion (blue) of total normalized *rpoB* (RNA polymerase) gene abundance (grey). The description of all marker genes and methane and nitrogen cycling processes is given in Table 1.

cluster and *Nitrosoarchaeum limnia* SFB1. The two other contigs clustered within an environmental group only distantly related to described MG-I AOA. This environmental cluster also contained the only contig that could be assembled from the OMZ core. A niche differentiation between shallow and deep-water clades of MG-I *Thaumarchaeota* has also been described before in the Arabian Sea OMZ *Villanueva, Schouten & Sinninghe Damsté (2014)*, but also in other marine environments (*Beman, Popp & Francis, 2008*; *Santoro, Casciotti & Francis, 2010*). Besides archaeal *amoA*, only a single read with low identity (<50% on the amino acid level) to known bacterial *amoA* reads was detected, indicating ammonium oxidizing bacteria (AOB) likely play only a small role in the Arabian sea OMZ although they were detected in other OMZs (*Molina et al., 2007*; *Lam et al., 2009*). Consistent with the absence of AOB *amoA*, only five reads matching *hao* were detected in the upper limit of the OMZ. In the core, 475 reads matched the *hao* database, but even after removal of the bonafide anammox hydrazine dehydrogenase hits (Fig. 3), over 90% of the *hao* matches were affiliated with *Scalindua*, which is known to encode up to ten paralogs of this protein (*Van de Vossenberg et al., 2013*; *Speth et al., 2015*).

The second step of complete nitrification, nitrite oxidation, is challenging to study using a marker gene approach as nitrite oxidoreductase (*NxrA*) and nitrate reductase (*NarG*) are homologous enzymes. A further complicating factor is the polyphyletic nature of the *nxrA* gene (*Lücker et al., 2010*). To account for this, we first extracted all the reads matching a

combined *narG*/*nxrA* reference set, and then used a second round of BLASTx and BLAST score ratio separation (threshold 0.85–0.95) to distinguish between *narG* and *nxrA*. No reads could be confidently assigned to *nxrA* of the *Nitrobacter/Nitrococcus/Nitrolancetus* group in either station. Conversely, 33% and 23% of the reads matching *narG*/*nxrA* were assigned to *nxrA* of the *Nitrospira/Nitrospina*/anammox group in PA2 and PA5, respectively. Further separation between *nxrA* matches, to distinguish the nitrifier *nxrA* from anammox, was achieved using iterative mapping. Classification using MEGAN indicated that anammox made up 44% of the reads matching *nxrA* at station PA2. As all other analyses indicated anammox bacteria were virtually absent from this station (Fig. 1 and Table S2), we explored these reads in detail using iterative mapping and manual curation of the sequences. This led to the identification of a novel lineage of *nxrA* clustering between anammox and *Nitrospina* sequences (Fig. 5), which was slightly more abundant than the retrieved *Nitrospina sp.* at both station PA2 (approx. 2% and 1.5% of the population respectively) and PA5 (approx. 3% and 2% of the population respectively). Interestingly, distinct lineages of both the putative novel nitrite oxidizer and *Nitrospina* seemed to occupy either station (Fig. 5). The abundance of anammox *nxrA* in the OMZ core sample correlates well with the abundance as assessed using the 16S rRNA gene (Table 1) and other anammox markers (Table S3, discussed below).

No other nitrite oxidizing organisms were detected. The detection of a significant abundance of nitrite oxidizers in OMZ ecosystems is consistent with a previous study showing that nitrite oxidation was an active process in the Namibian OMZ (*Füssel et al., 2012*).

### Genetic potential for processes contributing to nitrogen loss

Nitrite and nitrate resulting from nitrification can be readily used in denitrification, anammox and DNRA. To date, it is still unclear which process is dominating or if a combination of various processes occurs. Most of the recent studies indicated anammox instead of denitrification as prevalent pathway in OMZs (*Kuypers et al., 2005*; *Thamdrup et al., 2006*; *Hamersley et al., 2007*; *Lam et al., 2009*). However, two reports described high and active denitrification rates in the Arabian Sea OMZ (*Ward et al., 2009*; *Bulow et al., 2010*). In yet another study, Jensen and co-workers found that anammox coupled to DNRA was the prevalent process in this system (*Jensen et al., 2011*) and further studies confirmed a high abundance of anammox bacteria at the core of the Arabian Sea OMZ (*Pitcher et al., 2011*; *Villanueva et al., 2014*).

Here, we found that the nitrate reductase is by far the most dominant nitrogen cycle enzyme encoded in the Arabian Sea OMZ core (78% of normalized *rpoB* gene abundance, Fig. 4). This is consistent with observations in the Peruvian oxygen minimum zone (*Lam et al., 2009*; *Glass et al., 2015*) and in the Eastern Tropical South Pacific OMZ (*Stewart, Ulloa & DeLong, 2012*). The reduction of nitrate to nitrite is a crucial step as the nitrite forms the starting point for many subsequent processes: nitrite reduction in denitrification, in DNRA and in anammox. Additionally, nitrite can also be re-oxidized to nitrate. The genetic potential for all these processes is substantially encoded in the OMZ core. Consistent with Jensen and co-workers (*Jensen et al., 2011*), we find the potential for DNRA (*nrfA* encoding the penta-heme nitrite reductase as marker gene), approximately

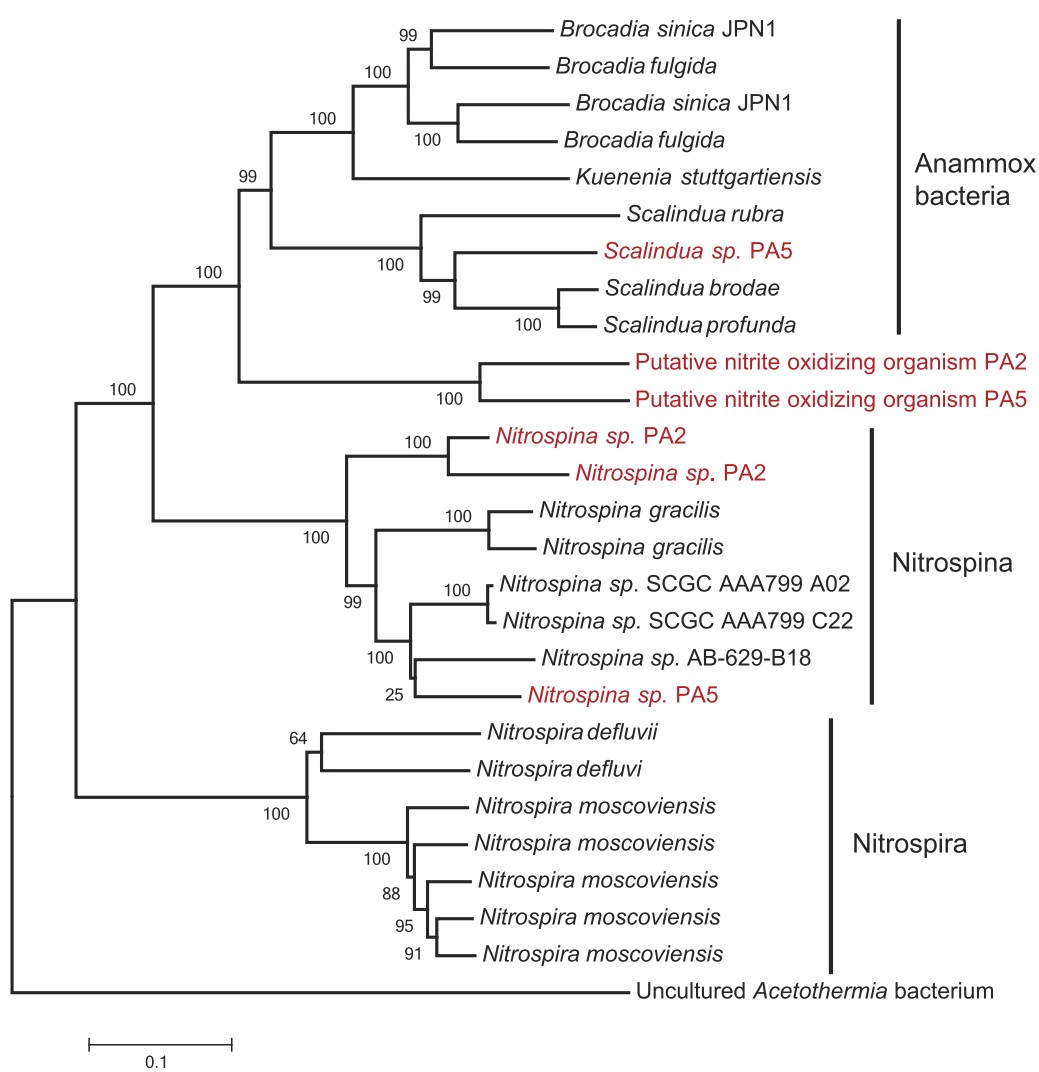

**Figure 5** **Phylogenetic inference of *nxrA* sequences from the OMZ metagenomes.** The tree was calculated using the Neighbor Joining algorithm and based on 3,209 nucleotide positions. Bootstrap values represent 1,000 replicates.

in equal abundance to anammox. Upon close inspection 60% of the reads matching *nrfA* originate from two closely related strains of an unknown organism. The 812 bp hybrid sequence obtained after assembly has 73% identity (AA level) to only two sequences in the database: *Coraliomargerita akajimensis* (*Verrucomicrobia*) and *Pelobacter carbinolicus* (*Deltaproteobacteria*). Although the phylogeny of the organisms most likely responsible for DNRA in the Arabian Sea OMZ remains unclear, retrieval of the divergent *nrfA* emphasizes the potential of our approach for novel microbiological gene discovery.

Only few sequences indicative for the process of denitrification were detected. Although many reads matched nitrite reductase encoded by either *nirS or nirK* (Table S3), 50% of all *nirS* reads could be classified as *Scalindua*-related. The *nirS* contig obtained after assembly of the reads matching *Scalindua* showed 77% nucleotide sequence identity to *Scalindua brodae* and 99% identity to unpublished sequences from the Gulf of California and

Eastern Tropical North Pacific OMZs (Genbank accession: KC596869). It corroborates the 16S rRNA gene analysis indicating the presence of one dominant uncultivated anammox ecotype. Of all detected *nirK* reads, 70% originated from AOA. These *Thaumarchaeota* are known to harbour multiple copies of *NirK*-like copper oxidases encoding genes (*Stahl & De la Torre, 2012*) explaining their high abundance in the datasets (Fig. 4). It has been previously suggested that nitrifier denitrification accounts for the majority of nitrous oxide observed in the ocean (*Babbin et al., 2015*; *Kozlowski et al., 2016*). Besides playing a role in nitrifier denitrification, these enzymes are also hypothesized to be the missing hydroxylamine oxidoreductase equivalent in AOA catalysing the oxidation of hydroxylamine to nitrite (*Stahl & De la Torre, 2012*).

Similar to the *nirS*-type nitrite reductase genes, we found that 40% of all reads matching the *norB/norZ* reference set (encoding the nitric oxide reductase) at both stations combined were affiliated with *Scalindua* (Table S3). Potential for the final step of denitrification via nitrous oxide reduction (encoded by the *nos* genes) was limited with 92 reads at station PA2 and only 27 reads at station PA5 (Fig. 4, Table S3). We could not find genomic indication for nitrogen loss via the nitric oxide dismutase of NC10 phylum bacteria as recently described for the Eastern Pacific OMZ (*Padilla et al., 2016*).

The abundance of anammox marker genes *hzsA* (encoding the hydrazine synthase) and *hdh* (encoding the hydrazine dehydrogenase) coincides well with the abundance estimates based on 16S rRNA, *nxrA*, and *nirS* genes consistently indicating that anammox bacteria of the *Scalindua* genus are present at approximately 5% abundance.

The *nifH* gene, encoding a subunit of the nitrogenase, was used as marker in screening for nitrogen fixation potential in the Arabian Sea datasets. Ocean circulation models have predicted highest nitrogen fixation rates close to zones of nitrogen loss (*Deutsch et al., 2007*) and were supported by recent studies reporting the presence and transcription of *nifH* genes in sub-oxic waters of the Arabian Sea (*Jayakumar et al., 2012*; *Bird & Wyman, 2013*). However, we detected no *nifH* hits in the PA2 metagenome and only 3 reads matched *nifH* in the PA5 dataset (Fig. 3). The *nifH* genes in the above mentioned studies were obtained using PCR amplification with specific primer sets able to detect a much lower abundance of diazotrophs in the environment. Our results indicate a low abundance of diazotrophs that nevertheless would easily be detected by PCR amplification. We can estimate the abundance of diazotrophs from our dataset, assuming an average microbial genome size of 3 Mbp and the *nifH* gene length of 900 bp. If present in all genomes, 3 reads per 10.000 should contain part of a *nifH*. However, only 3 reads were detected in the PA5 metagenome (1,6 million reads), which is 160 times lower and thus results in a diazotroph abundance estimate of 0.6% based on the dataset. PCR primers should be able to amplify genes present in organisms with this abundance. Alternatively, the diazotrophic community in our dataset was, for unknown reasons, lower than in other studies from similar ecosystems.

## Methane cycling potential

To examine the methane cycling potential in the Arabian Sea, we selected marker genes targeting methane production and methane oxidation (Table 1). The *mcrA* gene (encoding

the methyl-Coenzyme M reductase) was used as functional marker for methane production and anaerobic methane oxidation. We could not detect any *mcrA*-like sequences of canonical methanogens or sequences of the very recently described *mcrA* homologues from *Bathyarchaeota* (*Evans et al., 2015*). Also no sequences of anaerobic methane oxidizing archaea (ANME clades) were found. This is consistent with the absence of methanogen 16S rRNA genes and very low abundance (3 reads) of *Bathyarchaeota* 16S rRNA gene sequences. Thus, methanogenesis does not seem to play a major role in the Arabian Sea OMZ water column. Nevertheless, in marine ecosystems, a second pathway for methane production apart from methanogenesis was proposed (*Karl et al., 2008*). This aerobic pathway includes the cleavage of methylphosphonate (Mpn). Mpn was shown to be synthesized by the AOA *Nitrosopumilus maritimus* using the Mpn synthase as key enzyme and a homologue of this enzyme was also found to be encoded in SAR11 genomes (*Metcalf et al., 2012*). The cleavage of Mpn that results in the release of methane, is catalysed by the C-P lyase multi-enzyme complex (*Daughton, Cook & Alexander, 1979*). In *Escherichia coli*, the C-P cleavage is encoded in a 14 gene operon (*Metcalf & Wanner, 1993*). The composition of the gene cluster is variable among bacteria, however, the *phnGHIJKM* genes seem to be conserved and essential for activity (*Huang, Su & Xu, 2005*). We selected the *mpnS* gene (encoding the Mpn synthase) and the *phnGHI* genes (encoding components of the C-P lyase pathway) as functional marker genes for aerobic methane production in our datasets. In the PA2 dataset, 45 reads matching the *mpnS* gene were retrieved, most of which could be affiliated with the MG-I *Thaumarchaeota* (Fig. 6, Table S4). Comparing the *mpnS* read numbers with corresponding 16S rRNA gene and *amoA* read numbers from *Thaumarchaeota* results in a far lower abundance of *mpnS* reads in the dataset. However, this is in good agreement with the observation that not all MG-I *Thaumarchaeota* encode *mpnS* in their genome. The dominant MG-I ammonia oxidizer in the Arabian Sea datasets is closely related to *Nitrosopelagicus brevis* (Fig. 2), which does not possess the *mpnS* gene (*Santoro et al., 2015*). In the metagenome from the OMZ core zone, 7 *mpnS* reads (3 reads matching to MG-I) were detected.

The C-P lyase gene cluster has been found in many genomes of marine bacteria and could also be associated to growth with Mpn as sole phosphorous source (*Dyhrman et al., 2006*; *Martinez, Tyson & DeLong, 2010*; *White et al., 2010*). In the Arabian Sea datasets, no reads matching the *phnGHI* genes were detected in PA2 and only 3 *phnI* sequences affiliated to *Rhodobacteriales* (*Alphaproteobacteria*) were retrieved from PA5 (Fig. 6, Table S4). Depending on the dissolved inorganic phosphorous availability, the abundance of bacteria harbouring the gene cluster can vary between as much as 20% and below 1% of all bacteria (*Martinez, Tyson & DeLong, 2010*), indicating that the presence of the C-P lyase provide an advantage in phosphorous limited environments. We could not find a high abundance of this protein complex in the Arabian Sea OMZ, hence the ability for acquiring phosphorous from Mpn might be less important in this system.

In accordance with the absence of marker genes indicative for methane production, only few reads were retrieved that could be affiliated with methane oxidation. For aerobic methane oxidation, we used the marker genes *pmoA* and *mmoX* encoding the particulate and soluble methane monooxygenase. In the PA2 dataset, 15 *mmoX* and 3

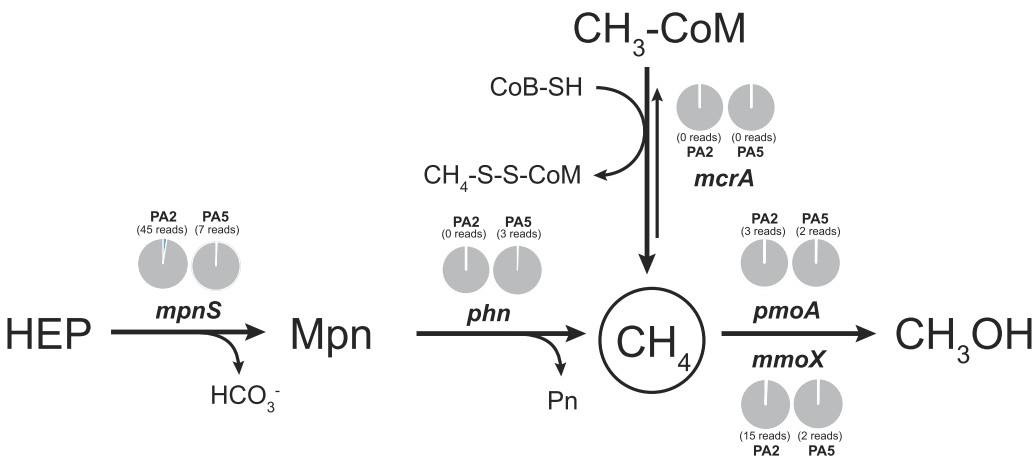

**Figure 6   Methane cycling potential in the Arabian Sea oxygen minimum zone.** Read abundances were normalized according to gene length and total read abundance in the metagenome dataset. Normalized abundances are shown as proportion (blue) of total normalized *rpoB* (RNA polymerase) gene abundance (grey). Original read abundances are given in addition below or above the pie charts. HEP: 2-hydroxyethylphosphonate. Mpn: Methylphosphonate. The description of all nitrogen and methane cycling marker genes is given in Table 1.

*pmoA*-like reads were detected. However, after closer inspection, all *mmoX* reads showed moderate sequence identity (up to 71% on amino acid level) to homologous toluene monooxygenases. For the *pmoA* reads, highest identity (68–97% on amino acid level) was found to the monooxygenase of the SAR324 clade (*deltaproteobacteria*). Based on 16S rRNA gene abundance, the SAR324 clade was found to belong, besides others, to the dominant bacteria in the Arabian Sea dataset (Fig. 1). So far, no enrichment of SAR324 clade bacteria is available and it is not known if this monooxygenase is used for methane or higher alkane oxidation. Whereas the *pmoA* phylogeny indicates relation to C2-C4 alkane monooxygenases (*Li et al., 2014*), the genomes of SAR324 members suggest the potential for both, C1 and higher alkane utilization (*Sheik, Jain & Dick, 2014*). Nevertheless, not all SAR324 genomes contain the alkane monooxygenase gene cluster (*Swan et al., 2011*). The PA5 dataset revealed 2 *pmoA* and 2 *mmoX*-like sequences, again only with moderate sequence identities to known alkane monooxygenases. Consistent with the absence of genes encoding the putative nitric oxide dismutase of NC10 phylum bacteria, we did not find any *pmoA* genes of these bacteria in the Arabian sea OMZ. Nevertheless, they might be present in a low abundance that could not be retrieved by our approach. Thus, if these bacteria play a role in methane and nitrogen cycling in OMZs, as suggested for the Eastern Pacific OMZ, needs to be investigated in future research (*Padilla et al., 2016*). In addition, 16S rRNA gene sequences of known aerobic methanotrophs were nearly absent. Only 3 reads clustering within gammaproteobacterial methanotrophs were present (1 read from PA2 and 2 reads from PA5). Although the overall abundance of hydrocarbon monooxygenase encoding reads is low in the Arabian Sea OMZ dataset, our analysis shows the existence of novel sequence diversity only moderately related to known sequences, that is not captured by currently used PCR primers.

## Conclusion

In this study, we compared the functional diversity in two metagenomes retrieved from the Arabian Sea oxygen minimum zone. Using manually curated reference databases, we screened the datasets for homologues indicative for nitrogen and methane turnover in this ecosystem. We are aware that the presence of genetic potential alone cannot be used to draw conclusions on activity of various processes. However, despite this limitation, the picture that emerges from our analysis is that the vast majority of organisms can contribute to nitrate reduction, probably coupled to degradation of organic matter and release of ammonium (*Kalvelage et al., 2015*). The nitrite formed by nitrate reduction can be re-oxidized to nitrate in a 'nitrite loop' ultimately resulting in removal of additional organic matter and release of more ammonium. The *nrf*-like nitrite reductase assembled from our dataset is only distantly related to described *nrf* sequences indicating that a yet unknown organism is responsible for this reaction in the Arabian Sea. The released ammonium can partially be oxidized by a diverse community of microaerophilic archaeal ammonium oxidizers, but in the core of the OMZ, the majority is likely converted by anammox bacteria, which contribute about 5% to the total abundance at the station PA5. Here, we could confirm the presence of a low diversity dominated by a Candidatus 'Scalindua arabica'-like anammox species as observed in previous studies (*Villanueva et al., 2014*; *Woebken et al., 2008*). Although denitrification was observed as the dominant process in another Arabian Sea study (*Ward et al., 2009*), our analysis, albeit only based on the genetic potential, does not support this point. Rather, an intricate nitrogen cycle involving many organisms and the exchange of intermediates and connection to other processes, as recently hypothesized for estuary and an aquifer (*Hug et al., 2016*, *Baker et al., 2015*) seems likely. The ultimate removal of nitrogen is most likely mediated by anammox. We found no evidence for methane turnover in the Arabian Sea OMZ, however, the metagenomes revealed the presence of new alkane monooxygenase diversity in this ecosystem.

## ACKNOWLEDGEMENTS

We thank the department of marine organic biogeochemistry and molecular lab facilities of the Royal NIOZ for support.

### Funding

Samples used in this manuscript were collected during the PASOM cruise funded by the Netherlands Organization for Scientific Research (NWO 817.01.015). This work was further supported by the European Research Council (ERC AG 339880 Eco_MoM and ERC AG 232937 anammox to MSMJ), the Dutch governments Gravitation grant (024002002 to MSMJ), and the Spinoza prize to MSMJ. CL and DRS were funded by BE-Basic FP 07.002.01. The funders had no role in study design, data collection and analysis, decision to publish, or preparation of the manuscript.

## Grant Disclosures

The following grant information was disclosed by the authors:
Netherlands Organization for Scientific Research: (NWO 817.01.015).
European Research Council: ERC AG 339880 Eco_MoM, ERC AG 232937.
Gravitation grant: 024002002.
BE-Basic grant: FP 07.002.01.

## Competing Interests

The authors declare that they have no competing interests.

## Author Contributions

- Claudia Lüke and Daan R. Speth conceived and designed the experiments, performed the experiments, analyzed the data, contributed reagents/materials/analysis tools, wrote the paper, prepared figures and/or tables, reviewed drafts of the paper.
- Martine A.R. Kox analyzed the data, contributed reagents/materials/analysis tools, reviewed drafts of the paper.
- Laura Villanueva conceived and designed the experiments, reviewed drafts of the paper, sampling.
- Mike S.M. Jetten conceived and designed the experiments, analyzed the data, reviewed drafts of the paper.

## Field Study Permissions

The following information was supplied relating to field study approvals (i.e., approving body and any reference numbers):

Samples used in this manuscript were collected in the Arabian Sea OMZ during the PASOM cruise funded by the Netherlands Organization for Scientific Research (NWO) under number 817.01.015.

## DNA Deposition

The following information was supplied regarding the deposition of DNA sequences:
NCBI Sequence Read Archive: SRA304624.

## Data Availability

Github: https://github.com/dspeth/bioinfo_scripts/tree/master/proteins.

## Supplemental Information

Supplemental information for this article can be found online at http://dx.doi.org/10.7717/peerj.1924#supplemental-information.

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
