# Peer review of "Metagenomic analysis of nitrogen and methane cycling in the Arabian Sea oxygen minimum zone"

_PeerJ, doi:10.7717/peerj.1924_

## Round 0.1 · original submission · Major Revisions

While both reviewers found that the described research presents a step forward in understanding microbial communities in OMZs, one of the reviewers expresses serious concerns as to how robust are the data and their interpretation. This is mainly due to the lack of detail in the description of methods, analytic and computational approaches in the current version. I think these concerns should be adequately addressed in a revised manuscript.

Reviewer 1 ·

Basic reporting

Please see "General Comments" section

Experimental design

Please see "General Comments" section

Validity of the findings

Please see "General Comments" section

Additional comments

Lüke et al. present a metagenomic analysis of two depths in the oxygen minimum zone (OMZ) of the Arabian Sea. The authors generate a custom database for homology searches to characterize the abundance and taxonomic diversity of marker genes of dissimilatory nitrogen and methane metabolism. These data are presented along with a taxonomic analysis based on 16S rRNA gene fragments recovered from the metagenomes. The resulting patterns are used to draw conclusions about the potential for nitrogen and methane transformations in a conceptual model of OMZ metabolism (Figure 6), similar to what has been done for other OMZ systems. The results constitute a comparatively minor advance in our overall understanding of OMZ microbial communities, as similar trends have been reported in other systems and few surprises are reported here. This may not have been the case had the analysis not been so restrictive in scope, essentially focusing only on a handful of target enzymes from the entire “meta” dataset. Nonetheless, the data will be of value for future comparative analyses demonstrating similarities (and differences) across OMZ systems, notably as this study provides one of the first metagenomic datasets from the Arabian Sea OMZ. The paper is generally well written (although see the caveat below) and should find an audience, assuming the following concerns can be addressed.

A) Methods lack clarity. There is a surprising and systemic lack of clarity and detail in the methods description. This makes it challenging to confirm the overall robustness of the results. A short list of examples is provided here:

1) Line 115 – What is meant by the “Arabian Sea depth profiles”? What data specifically?

2) Line 141 – What quality score cutoff was used for trimming?

3) Line 145 – What is meant by “high quality” here? Anything passing the initial screening? If so, it does not seem necessary to specify this again here (given line 142).

4) Line 152 – Classification by “visual inspection of the resulting phylogeny” is unclear. I.e., what was the threshold for categorizing a mapped read within a specific node? How were ambiguities resolved?

5) Database creation (line 154-166). Numerous issues here: a) This paragraph makes no mention of which genes were actually used as marker genes, and why. There is a cryptic reference to “Table 1” near the end of the paragraph (line 172) but more should be said here about marker gene choice. b) “sequences from public databases such as….” (Line 155) is ambiguous - were the named databases (Interpro, NCBI nr) used or weren’t they (were there others?)? c) Importantly, how were the downloaded sequences selected? Based purely on annotation? If so, at what level? I.e., were these sequences pulled from curated functional hierarchies (e.g., KEGG, pfam)? If not, what was the criterion for confirming the annotation? d) Was presence/absence of N’s the only quality criterion? e) How many sequences in total were downloaded? f ) Why was sequence selection for the norB/norZ database seemingly done differently? g) Does the “second step” mentioned in line 164 refer only to the norB/norZ analysis, or to the other enzymes as well? h) E-values are database-dependent – is it appropriate to use the same cutoff for both the searches against the custom databases and the searches against SILVA? i) The description of how BSR thresholds were determined for each enzyme (Lines 170-172) is not clear. Lines 268-277 provide some additional details (these details should be moved to the Methods) but the rationale for why this procedure is a valid approach for distinguishing false positives from novel/divergent variants needs to be discussed in more detail. I.e., it is not clear how empirical thresholds for distinguishing these data types were actually established. j) What parameters were used for assembly (line 176)? For that matter, why was an assembly of the total metagenomic dataset not performed (or was one performed? Lines 240-241 hint that this was done)?

6) How were the abundance estimates (read counts) normalized? Normalization is mentioned later in the text, but in a confusing way (i.e., line 277 mentions normalization to dataset size and gene length, while the Figure legends also discuss normalization to rpoB copies). Clarifying normalization procedures in the Methods would help interpret some of the conclusions drawn from the results. For example, it is not immediately obvious how the amoA counts are used to infer that Thaumarchaeota constitute 25% of the total community (line 285). Nor is it clear that the read counts presented throughout the discussion reflect standardization to a common dataset size (is this true?).

Overall, there is a critical lack of detail in the Methods description, notably regarding custom database creation and use. This needs to be corrected before the overall integrity of the results can be fully evaluated.

B) Sampling sites poorly described. For example, line 117 mentions the two stations, but says nothing about how they compare or how far apart they are. Are there any environmental differences between these sites that may affect how we interpret the resulting datasets as representative of the respective depth regimes? Furthermore, what explains the variation (gene content) between depths? Lines 186-187 reference dissolved O2 concentrations of 5 and 3.4 uM for the upper and lower depths, respectively. From a cursory perspective, 3.4 is not very different from 5 uM. What else is going on? A more comprehensive description of the physiochemical conditions at the sampling sites (not simply referencing Table S1) is necessary to help put the data into an environmental and comparative perspective.

C) Grammatical/spelling errors. The manuscript is replete with such errors, beginning in the first sentence of the abstract. Careful editing is required before publication, and should have been done before submission (this lack of attention risks bringing into question the overall thoroughness and integrity of the analysis).

D) Other (in order of appearance):

Abstract: It is unclear what “gene-centered” means here? Aren’t all metagenomic analyses “gene-centered”? Also, “comprehensive overview” is overstated given how few enzymes were actually analyzed?

Line 99: What are “high” methane concentrations? Please provide the actual value (or approximation).

Line 103. Is it really true that Thaumarchaeota are thought to be an “important” source of methane? It may be more appropriate to say “potential source,” given current data (or lack thereof) regarding production rates by this group.

Line 183 – “curated” is not appropriate here without additional clarity about how the databases were generated (see above)

Line 185 – “genetic potential for … carbon turnover” is not valid. Carbon metabolism was not the focus of this study.

Line 209-210 - It is not clear why proteorhodopsin is being discussed here.

Line 211 – “anoxic OMZ core”. Does 3.4 uM O2 constitute anoxia?

Line 212 – “attachment to particles”. Are the putatively aerobic organisms discussed here (e.g., SAR11, Thaumarchaeota, MG-II, etc) known to be typically particle-associated? If not, is attachment to particles a likely explanation for their presence in the OMZ? Could anything else be going on?

Line 233 – “similar presence” is vague. Please report the actual values.

Line 240-241 – These lines mention an assembly using the entire metagenomic datasets. Why wasn’t this discussed in the Methods?

Line 243 – Why were AOA and anammox microbes selected for more extensive 16S-based phylogenetic analyses? Why not other taxa as well? The rationale for this focus should be discussed.

Methane-related 16S sequences. Were 16S sequences from known methane-cycling taxa recovered in the analysis? This should be discussed given the focus of the paper.

Line 293 – Do the available Thaumarchaeota genomes (there are now several in public databases) contain multiple amoA copies? If so, it would be valuable to mention this here to help support the author’s interpretation of the amoA abundances.

Line 353-356 – “Nitrate reduction to nitrite conversion is the …. electron donor for nitrite oxidation” doesn’t make sense grammatically, though I think I understand the point.

Line 359 vs lines 366-367 (and elsewhere) – the text alternates between providing aa vs nt identity values for the different enzymes. For comparative purposes, it is better to be consistent.

Line 369 – The enzyme encoded by hdh is not specified here (not until line 383). Hdh should be defined/described when the gene name is first introduced. Please check to make sure the same is done for the other genes (e.g., the phn genes in line 416).

Line 395 – The calculation being made here is confusing. What does 0.6% relative abundance refer to? The abundance of nifH genes as a percentage of total genes? Of diazotroph genomes out of total genomes? Please specify.

Line 476 – “our analysis does not seem to confirm this point” is overly speculative without rate data.

Line 744 – Again, how is PA5 “anoxic” if there is 3.4 uM O2?

Reviewer 2 ·

Basic reporting

The paper “Gene-centric metagenomic analysis of nitrogen and methane cycling in the Arabian Sea oxygen minimum zone” by Claudia Lüke, Daan R. Speth, Martine A.R. Kox, Laura A. Villanueva, Mike S.M. Jetten, authors
used a gene-centric approach to compare the functional diversity in two metagenomes retrieved from metagenomes retrieved from the Arabian Sea oxygen minimum zone, an interesting zone where lost of nitrogen seems to be occur. It was found that the removal of nitrogen is most likely mediated by anammox.
They concluded that likely the intricate nitrogen cycle involving many organisms and the exchange of intermediates and connection to other processes. In addition to that authors explored the presence of genes involved in metabolism of methane and have found no evidence for methane turnover in the Arabian Sea OMZ. This study expanded our knowledge of the processes and biological diversity of this unique ecosystem.
The paper is well written, professional English language used throughout although with some exceptions (see comments and suggestions below).
Figures are relevant, high quality, well labelled & described.
The introduction previews the main points of the paper.
There is one suggestion for the structure of the paper (see comments and suggestions below). Overall , the paper is well structured.

Experimental design

Research questions well defined, relevant & meaningful. It is stated how research fills identified knowledge gaps. Data analyses performed to a high scientific and technical standards.

Validity of the findings

The data are robust, statistically and computationally sound.
Conclusions are appropriately stated.

Additional comments

The paper is well written, professional English language used throughout although with some exceptions (see comments and suggestions below).
Figures are relevant, high quality, well labelled & described.
The introduction previews the main points of the paper.
There is one suggestion for the structure of the paper (see comments and suggestions below).
Comments and suggestions for corrections:
1. 39- Thaumarchaota –typo - Thaumarchaeota
2. Why dissimilatory nitrite reduction abbreviated as DNRA?
3. 80 – probing instead of pairing
4. 99 – High methane concentration not saturation..
5. 121- “DNA was precipitated using ice-10 cold ethanol.” What is “ice-10 cold”?
6. 174-176 – “For selected genes, the reads were extracted and imported in the CLC genomics workbench for iterative mapping (used for assembly of nxrA, hdh) (Dutilh, Huynen & Strous, 2009) or direct assembly (used for hzsA, nrfA)”. Why two different methods were used for assembly of target genes?
7. 217 - Authors wrote “The deep branching bacterial phylum Marinimicrobia (formerly SAR406) (Fuhrman, Mccallum & Davis, 1993; Gordon & Giovannoni, 1996; Rinke et al., 2013) comprised 9% of all 16S rRNA affiliated sequence reads at station PA2. A recent transcriptome study indicates the involvement of these bacteria in extracellular proteolysis and fermentative amino acid degradation in a methanogenic environment (Nobu et al., 2015). These findings correlate well with their high abundance in a low oxygen environment (upper OMZ) and their dominance (20%) in the OMZ”. My question is: "How finding by Nobu et al made in methanogenic bioreactor can correlate well with presence of bacterial phylum Marinimicrobia in OMZ of Arabic see?"
8. Section “Genetic potential for nitrogen and methane cycling” in “Results and Discussion” should be moved to “Materials and Methods”
9. 353-355 - Sentence “Nitrate reduction to nitrite conversion is the starting point for subsequent nitrite reduction in denitrification, DNRA and anammox, as well as the electron donor for nitrite oxidation, all of which are substantially encoded in the OMZ core” Is not clear to me and should be corrected.
10. Also, 364 – Sentence “Potential for denitrification of nitrite to dinitrogen gas was limited in our datasets” needs to be reformulated. For instance: Small number of sequences which have been assigned to process of denitrification of nitrite to dinitrogen gas suggest low denitrification potential..
11. 410- typo “methyphosphotate”, should be methylphosphonate

---

## Round 0.2 · accepted · Accept

I am happy to inform you that the revisions to your manuscript were found to be satisfactory, and the manuscript can now be accepted. Congratulations.

Reviewer 1 ·

Basic reporting

see below, and prior review

Experimental design

see below, and prior review

Validity of the findings

see below, and prior review

Additional comments

The authors have done a thorough and commendable job of addressing my prior concerns, notably by greatly expanding the scope and clarity of the Methods. I look forward to seeing this paper in press.